# Grand canonically optimized grain boundary phases in hexagonal close-packed titanium

Enze Chen [1,2,3,4] ✉, Tae Wook Heo [2], Brandon C. Wood [2], Mark Asta [1,3] & Timofey Frolov [2] ✉

Grain boundaries (GBs) profoundly influence the properties and performance of materials, emphasizing the importance of understanding the GB structure and phase behavior. As recent computational studies have demonstrated the existence of multiple GB phases associated with varying the atomic density at the interface, we introduce a validated, open-source GRand canonical Interface Predictor (GRIP) tool that automates high-throughput, grand canonical optimization of GB structures. While previous studies of GB phases have almost exclusively focused on cubic systems, we demonstrate the utility of GRIP in an application to hexagonal close-packed titanium. We perform a systematic high-throughput exploration of tilt GBs in titanium and discover previously unreported structures and phase transitions. In low-angle boundaries, we demonstrate a coupling between point defect absorption and the change in the GB dislocation network topology due to GB phase transformations, which has important implications for the accommodation of radiation-induced defects.

Grain boundaries (GBs) are interfacial defects in crystalline materials that have long been studied for their influence on materials properties and performance[1]. Given their ability to exist in multiple stable and metastable states, which have been termed GB phases[2] or complexions[3], it is desirable to obtain an atomic-level understanding of the GB structures and possible phase transition pathways between them[4,5]. The GB phase transitions are believed to have a profound influence on an array of phenomena, such as diffusion[6] and GB migration[7] in materials.

Recent experiments have provided direct[8] and indirect[9] evidence for GB phase stability, coexistence, and transitions in metallic systems; however, given the vast five-dimensional space characterizing the macroscopic degrees of freedom (DOF) for GBs, it is not yet clear where these phases may appear. Atomistic simulations provide a powerful tool to guide such searches and unveil the microscopic mechanisms underlying the formation of GB phases[10]. Previous atomistic modeling studies have discovered a diverse array of GB phases present in face-centered cubic (FCC)[4,11,12], body-centered cubic

(BCC)[13,14], diamond cubic[15,16], and other cubic systems[17,18]. One notable feature shared by the aforementioned works is the ability to add or remove atoms from the GB region in the simulation cell, i.e., grand canonical optimization (GCO), which was required to access new ground states and metastable states. While the exchange of atoms at an interface could naturally occur in real polycrystalline materials due to diffusion, irradiation, and mechanical deformation at finite temperature, this variation is omitted in the majority of computational simulations employing the $\gamma$-surface method[19]. The $\gamma$-surface method is the traditional technique for simulating GBs where only relative translations are allowed between two bulk slabs before the atoms are relaxed using conjugate gradient energy minimization to their equilibrium positions at 0 K. It is often adopted for its simplicity, but the deficiencies exposed by the previous studies suggest that more DOF must be considered during optimization in order to find the true ground-state structure in certain GBs. A few alternative approaches from the literature for atomistic modeling of GBs include $\gamma$-surface with atom deletion[20], vacancy loading[21], high-temperature molecular

[1]Department of Materials Science and Engineering, University of California, Berkeley, CA, USA. [2]Materials Science Division, Lawrence Livermore National Laboratory, Livermore, CA, USA. [3]Materials Sciences Division, Lawrence Berkeley National Laboratory, Berkeley, CA, USA. [4]Present address: Department of Materials Science and Engineering, Stanford University, Stanford, CA, USA. ✉e-mail: enze@stanford.edu; frolov2@llnl.gov

dynamics (MD) simulations[4,15,22], Monte Carlo sampling[23,24], and evolutionary algorithms[11,17,24–26] that can access a greater diversity of structures and atomic densities at the interface, with a concomitant trade-off in computational complexity. These algorithms have optimized GB structures in a variety of systems, although seldom in a high-throughput manner[22,23], and it remains unclear if the ubiquity of GB phases that they have yielded extends to lower-symmetry crystalline systems that are common in nature and engineering applications.

A particular system of immense technological relevance is the hexagonal close-packed (HCP) crystal structure, which is considerably more complex than cubic systems, as it displays anisotropy in its crystalline lattice vectors and has a basis containing more than one atom. The HCP structure is adopted by elemental metals such as Mg, Zr, and Ti, the last of which ($\alpha$-Ti) will be the focus of this work. Ti alloys are important structural alloys for aerospace, biomedical, and energy applications, particularly where high specific strength and strong corrosion resistance are desired[27]. The importance of GBs in the $\alpha$-Ti system is highlighted in recent studies that used grain refinement to mitigate low-temperature oxygen embrittlement in $\alpha$-Ti[28] and 3D electron backscatter diffraction to map out the complete GB character distribution in this system[29]. In a previous study[30], we used an evolutionary algorithm to discover a ground-state structure for the $\{11\bar{2}4\}[1\bar{1}00]$ twin boundary (TB) in $\alpha$-Ti that was in closer agreement to density functional theory (DFT) calculations and high-resolution transmission electron microscopy results than previously reported structures. In addition to these experimental works, there are also several atomistic simulation studies in the literature that systematically model symmetric tilt grain boundaries (STGBs) in $\alpha$-Ti along the [0001][31,32], $[1\bar{1}00]$[33–36], and $[1\bar{2}10]$[36,37] tilt axes. Despite the simplicity of STGBs—only a tilt axis and tilt angle ($2\theta$) are required to describe the crystallographic misorientation between two bulk crystals—they include coherent TBs as an important subclass, several of which are experimentally observed in deformation microstructures and thus important for mechanical behavior in $\alpha$-Ti[27,28,30]. STGBs are also model systems to study the geometric relationships of defects at the interface[37]; however, as the previous studies utilized the $\gamma$-surface method, it is important to clarify the effects of GCO on STGB structure in $\alpha$-Ti and more broadly whether interfacial phases exist in HCP metals.

Herein, we perform GCO of low-index STGBs in $\alpha$-Ti using an open-source GRand canonical Interface Predictor (GRIP) tool that we developed to rigorously sample microscopic DOF at the GB. We use this tool along with empirical potentials to perform GB structure search, discovering new ground-state structures and GB phases. We further employ high-temperature MD simulations to explore the $\{2\bar{1}\bar{3}0\}[0001]$ STGB and demonstrate GB phase [meta]stability and phase transitions through a dislocation-pairing mechanism. We conclude by discussing the broader implications of these results on GB phase behavior in HCP systems and how the GRIP tool can benefit future studies for diverse crystal structures.

## Results

### Grand canonical optimization—the GRIP tool

GB structure prediction is a long-standing challenge in materials modeling that requires rigorous and often advanced sampling of possible interfacial structures. Previous studies of GBs in HCP metals generated the interfaces using the common $\gamma$-surface method[19], which is not guaranteed to yield the true ground state configuration in general[11,15]. In the traditional approach, conjugate gradient minimization from different starting points representing distinct relative translations of the grains across the boundary simply allows the atoms to fall into a nearby local minimum, which may be far away from the ground state. For example, complex GB core configurations may exist that require significant rearrangement of the constituent atoms[11,15].

The other significant limitation of the $\gamma$-surface method is that it is not grand canonical: All GBs created using this method are composed of the same number of atoms derived from the constituent perfect half-crystals. This poses a substantial constraint because many other structures, including true ground states, can be realized out of a different number of atoms at the interface[4,13,15]. For STGBs and a fixed reconstruction area, the number of distinct atomic densities that can give rise to different GB structures is given by the total number of atoms in one atomic plane parallel to the boundary, which we denote $N_{plane}^{bulk}$. This quantity is the limit because removing a full plane of atoms from a crystal will return the exact same configuration up to a relative grain translation.

Here we address these shortcomings through the development of an open-source tool GRIP to perform grand canonical GB structure search. During the optimization, we systematically explore all possible microscopic DOF by sampling different relative grain translations and atomic densities (see Methods for details). The latter is accomplished by randomly removing a fraction of atoms between 0 and $N_{plane}^{bulk}$ from the boundary plane. For a fixed translation and number of GB atoms, we optimize the GB structure using dynamic sampling (performed here using MD simulations) at different temperatures within a wide window between room temperature and 1200 K (approximately $T_{\alpha \to \beta}$ for Ti), and for different durations up to 0.6 ns. At the end of each MD run, we perform conjugate gradient energy minimization at 0 K until convergence before calculating the GB energy, $E_{gb}$ (see Eq. (1) in Methods). While it is well known that the quench rate impacts the success rate of generating the low-energy state[15], this dependence is largely obviated when the equilibrium GB structure forms during the finite temperature simulations, from which a rapid quench is expected to yield the low-energy state. The random and diverse GB structure initialization coupled with the extensive dynamic sampling for each microscopic DOF done by hundreds of parallel calculations ensures a rigorous GB structure exploration.

To further underscore the need for rigorous sampling, Fig. 1 shows the structural diversity and success rates from randomly sampling MD simulation parameters, namely temperature ($T$) and duration (MD steps). Figure 1a shows the range of $E_{gb}$ as a function of $T$ when the duration is fixed, and only at intermediate temperatures does the algorithm find the ground-state structure. For this representative boundary, the U shape of the $E_{gb}$ vs. $T$ plot illustrates the inefficient frozen dynamics of the GB structure at low temperatures and the generation of disordered liquid-like GBs at very high temperatures. From this data, we can compute the probability of finding the ground state—calculated as the fraction of ground-state structures out of all sampled structures at each $T$—which peaks at approximately 1200 K and is zero for very low and very high temperatures (Fig. 1b). These panels illustrate the existence of an optimal temperature range that is sensitive to the structural DOF of each system and outside of which the GB may fail to be optimized. Analogously, simply choosing an optimal $T$ (e.g., 1000 K) is insufficient, as too few MD steps will never achieve the ground state, as shown in Fig. 1c. The energy range, $\Delta E_{gb}$, is plotted in Fig. 1d to show how the spread generally decreases as the MD duration increases; however, we emphasize that the optimal parameters are not known a priori. These optimal parameters can vary significantly not only with the boundary character described by the five macroscopic DOF, but also for larger area reconstructions of the same boundary. GRIP performs a more thorough sampling of structures and optimization parameters compared to regular MD simulations with open surfaces. The periodic boundary conditions (PBCs) allow for convenient calculation of GB energy and atomic density, thereby enabling robust, high-throughput optimization of large GB datasets.

As motivated in the Introduction, we showcase the performance of GRIP in the following sections through a detailed analysis of STGBs in HCP $\alpha$-Ti. Importantly, however, we note that we also comprehensively benchmark our tool by reproducing well-studied literature

results for tilt and twist GBs in elemental cubic metals[4,14] and more challenging covalently-bonded, lower-symmetry systems[15,23] (Supplementary Fig. 1). Even in the thoroughly studied BCC W system[14], we discover a ground-state structure with a different GB atomic density and markedly different dislocation network in the GB than previous reports (Supplementary Fig. 2). Such results, while not discussed further in this work, underscore the opportunities of having a robust method for exploring GB phase space across disparate chemical systems. These discoveries position GRIP as a tool capable of advancing the state-of-the-art in GB structure prediction through its extensive dynamic sampling of the relevant DOF.

### Survey of GB phases in HCP $\alpha$-Ti

The two-atom basis of HCP Ti presents additional considerations during optimization, and Fig. 2 illustrates one nuance in having two possible cases of calculating $N_{plane}^{bulk}$. For the orientation shown in Fig. 2a, all atoms found inside the planar region have the same $z$-positions indicated by the dashed magenta line. Such orientations are analogous to cubic systems and have only one distinct surface termination. For the second case shown in Fig. 2b, the atoms belonging to

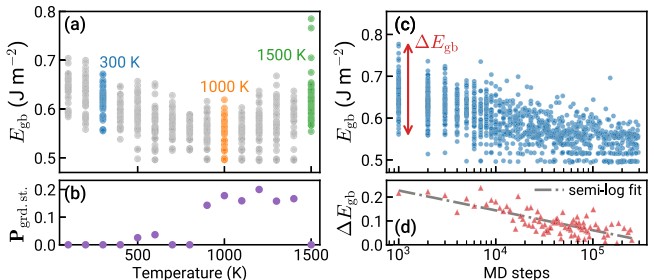

**Fig. 1 | Success rate as a function of search parameters in the Grand canonical Interface Predictor (GRIP).** The success rate of finding the ground state is sensitive to the search parameters, including temperature and duration of the molecular dynamics (MD) sampling. The optimal parameters are not known a priori and vary for each particular grain boundary (GB) and reconstruction, demonstrating why rigorous sampling of the parameter space is critical. **a** We plot the GB energy, $E_{gb}$, against the temperature, $T$, for a representative GB search, where each point is a single iteration of GRIP. The U shape of the plot illustrates the inefficient frozen dynamics of the GB structure at low $T$ and the generation of disordered liquid-like GBs at very high $T$. **b** The fraction of ground-state structures ($P_{grd.st.}$) out of all structures sampled at each temperature are plotted. **c** We plot $E_{gb}$ against the number of MD steps, where each point is a single iteration. A sufficiently large number of MD steps is required to obtain the ground state, even at an optimal $T$. **d** $\Delta E_{gb}$ is plotted against the number of MD steps. The least-squares regression line shows the general convergence in energy at longer duration.

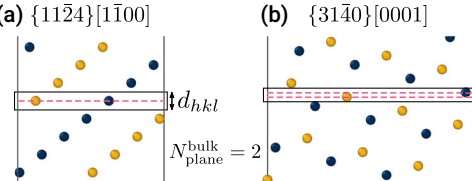

**Fig. 2 | Calculation of the number of atoms per plane, $N_{plane}^{bulk}$, in hexagonal close-packed (HCP) crystals.** The accurate calculation of $N_{plane}^{bulk}$ ensures that grain boundary structures with all possible atomic densities are explored. Because HCP crystals have two basis atoms, two different cases are possible when (**a**) all atoms inside the plane have the same $z$-position, or (**b**) they have two distinct $z$-positions resulting in two structurally different surface terminations. In both cases, $N_{plane}^{bulk}$ is calculated as the total number of atoms found inside the boxed region spanned by the hexagonal interplanar spacing, $d_{hkl}$ (calculated using Eq. (3) in Methods). The distinct $z$-positions of atoms belonging to the same plane are indicated by the dashed magenta lines.

the same plane can have two distinct $z$-positions, giving rise to two structurally different surface terminations. These two distinct terminations are possible because HCP has two basis atoms. In all orientations, the thickness of the planar region, $d_{hkl}$, corresponds to the smallest normal component of a lattice vector connecting two atoms on the same sublattice with different $z$-positions. We further note that this definition of a plane of atoms works for both cases, allowing us to uniformly apply it in calculating GB atomic density, [$n$] (see Eq. (2) in Methods).

Figure 3 shows the results of the GRIP searches for two representative boundaries evaluated using a modified embedded-atom method (MEAM) potential[38], illustrating the need for grand canonical structure optimization for GBs in HCP Ti. Panels (a) and (d) show plots of $E_{gb}$ vs. [$n$], which was introduced for cubic crystals in our previous work[4,11]. Each point on the plot corresponds to a particular GB structure obtained after energy minimization. The thorough exploration enabled by GRIP generates hundreds of distinct structures covering different densities and energies.

For the $\{11\,2\,\overline{13}\,0\}[0001]$ GB, the structure search identifies two GB phases with different atomic densities [$n$] = 0 and [$n$] = 0.75. The structures are illustrated in panels (b) and (c), respectively. The [$n$] = 0 phase does not require insertion or removal of atoms and is metastable at 0 K. It is composed of well-separated cores of edge dislocations with Burgers vector $\mathbf{b}_1 = \frac{1}{3}\langle 1\overline{2}10 \rangle$, as identified in green by the dislocation extraction algorithm (DXA) in OVITO[39,40]. The predicted ground state of this boundary has [$n$] = 0.75 and thus cannot be generated by using the simplistic $\gamma$-surface approach or sampling different terminations. Its structure is significantly different from the [$n$] = 0 state, where the dislocation cores overlap and the boundary structure appears completely flat. The energy of the ground state (corresponding to [$n$] = 0.75) is 1% lower than that of the metastable phase ([$n$] = 0). We perform additional MD simulations at high temperature (up to $T$ = 1000 K) for 20 ns with PBCs and observe that both structures remain stable for the entire duration. We finally note that this GB is a single-surface termination type of boundary where all atoms belonging to a bulk plane have the same $z$-coordinate.

The GRIP search for the $\{31\overline{4}0\}[0001]$ GB shown in Fig. 3d illustrates that sampling the microscopic descriptor [$n$] allows GRIP to identify all relevant distinct GB configurations, even for orientations with two distinct surface terminations. Similar to the previous boundary, the prediction of the ground-state structure at [$n$] = 0.5 also requires an insertion (or removal) of half of the atoms in one $\{31\overline{4}0\}$ plane; however, different from the first example, this particular ground-state structure can also be generated using the $\gamma$-surface approach that considers two possible surface terminations. The different surface terminations are obtained in a straightforward manner by removing half of a plane that contains two layers of atoms, as visualized in Supplementary Fig. 3. We emphasize that while sampling terminations may suffice in some cases, it is clearly restricted to atomic densities of 0 and 0.5, thereby performing very limited optimization of the atomic structure. In our search, for example, the GRIP tool finds a GB structure at [$n$] = 0 with $E_{gb}$ = 0.509 J m$^{-2}$ (green circle), approximately 18% lower in energy than the best $\gamma$-surface structure (blue triangle). For comparison, the ground-state structure and the two metastable states at [$n$] = 0 are shown in panels (e), (f), and (g).

Our structure searches performed for 150 GBs with three different tilt axes show that the need for GCO and presence of multiple GB phases is a general phenomenon in HCP Ti. Figure 4 summarizes the results from GRIP for the family of [0001] STGBs studied. Each subplot is equivalent to the gray boundary in Fig. 3a, denoting the minimum-energy structures at different [$n$] and the square marks the ground state. Evidently as many of the minima are located at [$n$] = 0.5, GCO is necessary to find the ground state in multiple [0001] STGBs. Similar to $\{31\overline{4}0\}$, the $\gamma$-surface method often performs poorly for these GBs, getting higher energies and different structures than the GRIP tool.

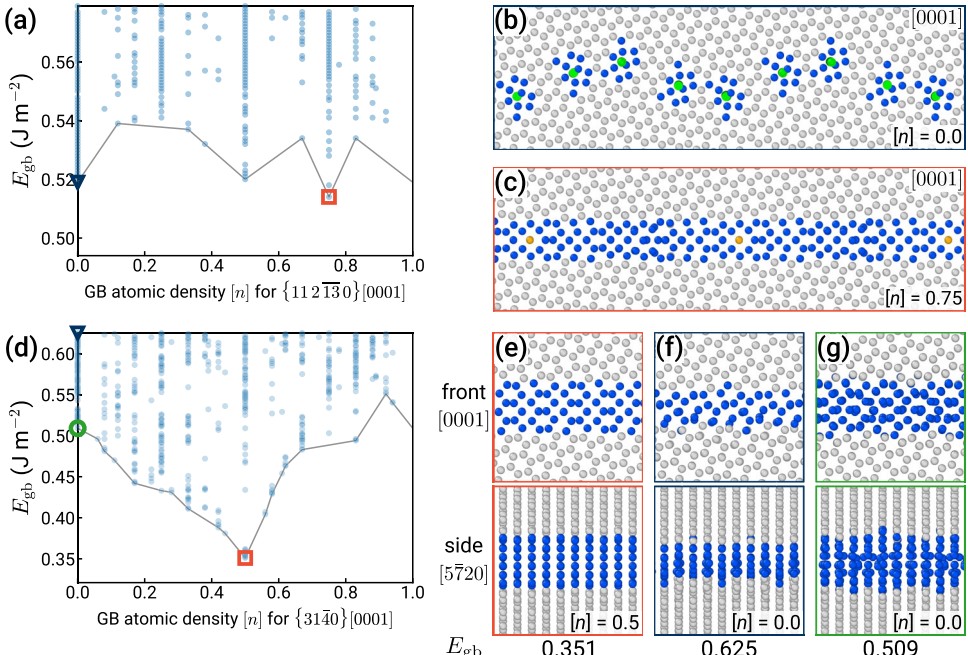

**Fig. 3 | Representative grain boundary (GB) structure searches using the Grand canonical Interface Predictor (GRIP). a** GB energy $E_{gb}$ vs. GB atomic density $[n]$, where each point is a distinct structure. The plot reveals two GB phases of $\Sigma49\{11\,2\,\overline{13}\,0\}[0001]$, marked with a blue triangle and red square, which are shown in (**b**) and (**c**), respectively. **b** The $[n] = 0$ GB phase, composed of edge dislocations, is metastable at 0 K. **c** The $[n] = 0.75$ GB phase is the ground state, which requires grand canonical optimization. **d** $E_{gb}$ vs. $[n]$ for $\Sigma13\{31\overline{4}0\}[0001]$. On the right, two orthogonal projections are shown for each minimum-energy structure obtained using (**e**) GRIP at $[n] = 0.5$ (red square), (**f**) the $\gamma$-surface method (blue triangle), and (**g**) GRIP at $[n] = 0$ (green circle). The atoms are colored according to the common neighbor analysis in OVITO[40,70].

The color map of the ground states reveals three distinct intervals that correspond to different GB structural units. The low-angle GBs in the intervals $\theta \leq 6.58°$ and $\theta \geq 23.41°$ are composed of isolated $\mathbf{b}_I$ edge dislocations (green markers) that for the lowest angles do not require GCO. The near-energy-degenerate minima at $[n] = 0.5$ are composed of the same type of dislocations, with the extra atoms accommodated by dislocation climb, resulting in GB structures with unevenly spaced GB dislocations. Different GB dislocations stabilize at $\theta \approx 10.89°$ with twice the Burgers vector of $\mathbf{b}_{II} = \frac{1}{3}\langle2\overline{4}20\rangle$ (purple markers). We investigate the transition between these two states in this GB in detail in the next section. For high-angle GBs in the interval $13.90° \leq \theta \leq 21.79°$, the ground states at $[n] = 0.5$ are composed of structural units that match the dislocation core structures of $\mathbf{b}_{II}$, as outlined in Supplementary Fig. 4. Additional energy maps for select $[1\overline{1}00]$ and $[1\overline{2}10]$ STGBs with low and high tilt angles out of 134 total studied are shown in Fig. 5, and results for the embedded-atom method (EAM) potential[41] are presented in Supplementary Figs. 5 and 6, and Supplementary Note 1. We caution that a reliable interatomic potential (IAP) is crucial for accurate structure prediction; moreover, energy differences between different IAPs may be much larger than the energy differences between distinct GB phases[42] (evident in Supplementary Fig. 11). We also perform select DFT calculations using the optimized GRIP structures as inputs to confirm the stability of the GB dislocation core structures and the relative energies between phases (see Supplementary Fig. 7). Taken together, these results demonstrate the ubiquitous need for grand canonical sampling in locating the ground-state GB structures for multiple tilt axes in HCP Ti.

## Phase transitions and coexistence

The multiple GB phases predicted by GRIP opens up an opportunity to explore GB phase transformations in HCP Ti; specifically, we focus on low-angle STGBs and investigate transformations that change the topology of the dislocation network arrangement. By elucidating the transformation pathways, we are able to predict the structure of a nucleus with a distinct dislocation network topology embedded inside a different parent dislocation network. We use point defects to drive the transformation and we study the coupling between defect absorption and changes in the dislocation network topology. While low-angle GB phase transformations due to solutes and temperature have been previously reported by experimental observations and simulations in a few metals[8,14,43–45], a clear connection between intrinsic point defects and distinct, equilibrium, low-angle GB phases remains missing.

We select the $\{21\overline{3}0\}[0001]$ STGB, which marks the transition between the two different GB dislocation types at a misorientation angle of $\theta \approx 10.9°$. The structure search performed on this GB is illustrated in Fig. 6, where we identify two distinct GB phases corresponding to atomic densities $[n] = 0$ (green circle) and $[n] = 0.5$ (orange square). Both structures correspond to GB energy cusps with respect to $[n]$ and the structures of the two phases at 0 K are shown in Fig. 6b and c, respectively. The ground state ($[n] = 0$) is composed of an array of $\mathbf{b}_I = \frac{1}{3}\langle1\overline{2}10\rangle$ edge dislocations, while the second phase is composed of $\mathbf{b}_{II} = \frac{1}{3}\langle2\overline{4}20\rangle$ edge dislocations with Burgers vector twice that of the ground state and consequently half the line density within the GB plane. The optimized dislocation core structures are consistent with the 'T' and 'A' structural units, respectively, reported by Wang and Ye using constrained molecular statics[31]. We perform MD simulations of each structure at temperatures as high as 1150 K for up to 20 ns to confirm that they are dynamically stable and indeed represent two GB phases. The other two energy cusps at $[n] = 0.33$ and $[n] = 0.67$ are the mixed states expected from the lever rule, where the GB region is patterned by weighted fractions of $\mathbf{b}_I$ and $\mathbf{b}_{II}$ dislocations corresponding to the proportions between $[n] = 0$ and $[n] = 0.5$, as shown in Supplementary Fig. 8.

Because the two GB phases are composed of different numbers of atoms, first-order transitions between the two structures can be triggered by changing the concentration of point defects[4]. The requisite high, local non-equilibrium concentrations of vacancies or interstitials

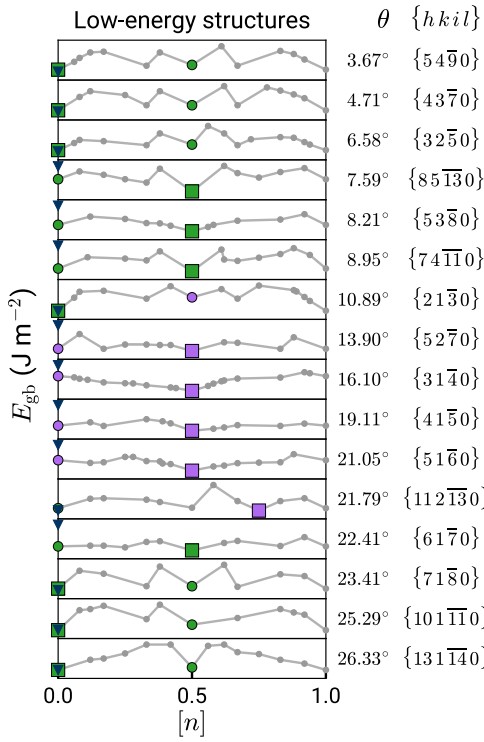

**Fig. 4 | Grain boundary (GB) energy map of [0001] symmetric tilt GBs.** The algorithm finds unexplored ground states and multiple GB phases within the entire misorientation range. Each subplot is analogous to the boundary of the plot of energy $E_{gb}$ vs. density $[n]$ in Fig. 3a, marking the minimum-energy structures at different GB atomic densities. The squares mark the ground state for each tilt angle, the larger circles mark a metastable state, and the blue triangles mark the $\gamma$-surface structure. The green and purple marker colors correspond to different GB phases (commensurate with the dislocation core colors in Fig. 6). Also shown are the tilt angles $\theta$ and Miller–Bravais indices {$hkil$}.

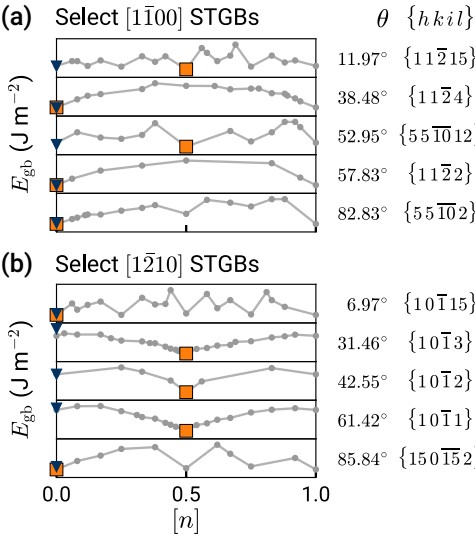

**Fig. 5 | Additional energy maps of symmetric tilt grain boundaries (STGBs) in titanium.** Five different misorientations are selected for the (**a**) $1\bar{1}00$ tilt axis and (**b**) $1\bar{2}10$ tilt axis as representative boundaries. Minima (indicated with orange squares) at $[n] = 0.5$ indicate that grand canonical optimization is broadly required to find the ground states in all families of STGBs studied in $\alpha$-Ti. Also shown are the tilt angles $\theta$ and Miller–Bravais indices {$hkil$}.

may occur, for example, as a result of radiation damage, rapid cooling from high temperatures, or deformation by creep. To mimic these conditions, we insert extra atoms into interstitial sites in the ground-state structure, triggering a local transformation of the GB structure illustrated in Fig. 7. During the transformation, the $\mathbf{b}_I$ dislocations of the ground-state structure pair up into $\mathbf{b}_{II}$ dislocations and absorb the extra atoms. Analogously, adding Ti atoms to the left half of the metastable structure and performing high-temperature MD triggers a dislocation-unpairing transition ($\mathbf{b}_{II} \rightarrow 2\mathbf{b}_I$) as shown in Fig. 7b. The transformed states remain stable at finite temperature and the transformation can be reversed by introducing vacancies near the GB, which we show in Supplementary Fig. 9. Effectively, this sequence of states and partial transformations illustrate the possibility of GB transformation-mediated creep[46,47]. Indeed, such a bicrystal can grow (shrink) by periodically alternating its GB structure and absorbing only half a plane of atoms (vacancies) at a time. If only one GB phase were present, the whole plane of atoms would have to be absorbed in concert through disconnection motion before returning to the original GB structure.

The simulated heterogeneous states containing two different GB phases show stable coexistence in the closed system at high temperatures. While not visible in Fig. 7a, b, the two phases are separated by a line defect called a GB phase junction, which is a dislocation as well as a force monopole[48]. The Burgers vector of this junction is non-zero because the GB phases have different dimensions[49]. The structure of this defect becomes more apparent when considering nucleation in fully 3D. To illustrate the shape of the nucleus during such a transformation, we increase the cross-section of the GB and place interstitial atoms of Ti (light blue) in a relatively small section. During the

subsequent high-temperature simulation at $T = 1000\,\text{K}$, the extra atoms diffuse to the boundary core and locally trigger the pairing transition. The equilibrium structure of the obtained nucleus is illustrated in Fig. 7c. The transformation changes the dislocation network topology as the dislocations of the parent structure shown in green ($\mathbf{b}_I$) pair up on the nucleus boundary to form three individual purple segments ($\mathbf{b}_{II}$). GB phase nucleation by absorption of point defects has been previously investigated in high-angle boundaries[4,14]. The important distinction of the transformation studied here is that it occurs in a low-angle GB; therefore, the core structure of the GB phase junction is represented by a collection of dislocation nodes where two dislocations pair up into one.

## Discussion

In this work, we perform grand canonical GB structure search to discover several GB phases in an HCP metal, $\alpha$-Ti. While GBs in $\alpha$-Ti have been investigated extensively[31–37], prior simulations were restricted to a fixed number of atoms derived from perfect surface terminations with no point defects (see Supplementary Note 1). By rigorously exploring atomic densities at GBs, we show that the minimum-energy structures can be found for atomic densities inaccessible to the $\gamma$-surface method for both high-angle and low-angle GBs across the misorientation range. The ubiquitous need for GCO and presence of multiple GB phases with different atomic densities is consistent with phenomena previously illustrated in elemental cubic metals with FCC[4,11,12] and BCC[13,14] crystal structures.

Subsequent high-temperature MD simulations guided by this detailed sampling of phase space yield the discovery of an unexpected GB phase transformation mechanism. The two phases shown in Fig. 6 are composed of periodic arrays of edge dislocations with distinct localized cores that contain different atomic densities in the GB. In the transition between these phases, dislocations of a less dense GB ($\mathbf{b}_I$, $[n] = 0$) pair up to form a new dislocation core ($\mathbf{b}_{II}$, $[n] = 0.5$), leading to a doubling of the Burgers vector and absorption of interstitial atoms. Previous studies of GB phase transitions in low-angle GBs revealed defect absorption by individual dislocation cores without the change of the Burgers vector[11,14] and other studies demonstrated the change in the dislocation network topology due to temperature[50], interstitial

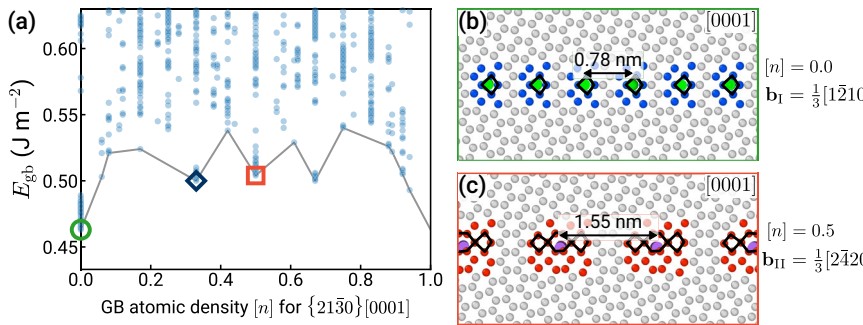

**Fig. 6 | Grain boundary (GB) phases of Σ7{2$\bar{1}$$\bar{3}$0}[0001]. a** The plot of GB energy $E_{gb}$ vs. atomic density [n] reveals two GB phases. The ground state at [n] = 0 and a metastable phase at [n] = 0.5 are shown in (**b**) and (**c**), respectively. Both states are composed of edge dislocations indicated by green and purple lines (identified using the dislocation extraction algorithm in OVITO[39,40]). The Burgers vector of the metastable boundary is twice that of the ground state. The non-hexagonal close-packed atoms of the dislocation cores are colored according to the common neighbor analysis[70] and structural units are outlined in black in (**b**) and (**c**).

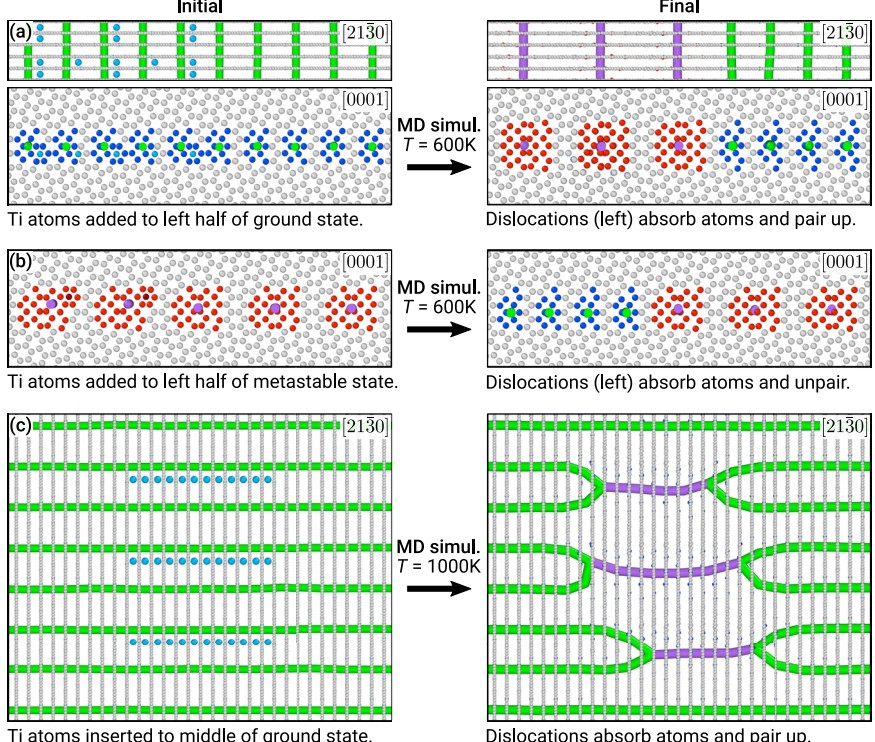

**Fig. 7 | Topological grain boundary (GB) dislocation network transformation in {2$\bar{1}$$\bar{3}$0}[0001]. a** Adding Ti atoms (light blue) to the left half of the ground-state structure and performing high-temperature molecular dynamics (MD) simulations triggers a dislocation-pairing transformation $2\mathbf{b}_I \rightarrow \mathbf{b}_{II}$ in a quasi-2D geometry. The top panels clearly show how the green dislocation lines pair up to form purple ones. The gray atoms are hexagonal close-packed coordinated while red and dark blue atoms highlight different dislocation core structures. **b** Analogously, adding Ti atoms (dark red) to the left half of the metastable structure triggers a dislocation-unpairing transition locally, $\mathbf{b}_{II} \rightarrow 2\mathbf{b}_I$. **c** Topological transition of the GB dislocation network upon defect absorption. The view of the GB plane shows a paired-dislocation GB island (nucleus in purple) inside the parent ground state (green dislocation lines).

absorption[44,45], and solute segregation[43]. It is also well established and expected that individual dislocations absorb point defects by climb[51]; yet here, we demonstrate a different mechanism where the dislocation network topology and the number of constituent atoms are coupled through distinct, equilibrium GB phases. This coupling suggests an important mechanism for point-defect absorption in polycrystalline materials with non-equilibrium concentrations of point defects produced by rapid quenching, irradiation, or additive manufacturing approaches that can yield dense dislocation cellular walls[52]. This work thus provides important insights into the ways in which low-angle GBs and dislocation arrays interact with point defects[44,45,53]. Additionally, we use high-temperature MD simulations with open surfaces to demonstrate first-order structural transformations between the

different GB phases (Supplementary Fig. 10). The work here, focused on an HCP metal, may be particularly relevant for engineering materials with such structures that experience radiation damage, such as Zr-based nuclear fuel cladding[54].

Herein, we further extend the notion of the number of atoms in a GB plane ($N_{plane}^{bulk}$) to non-cubic, multi-basis crystals like HCP. Previous studies on elemental cubic metals calculated this quantity as the total number of atoms located in one planar cut parallel to the GB, i.e., all these atoms are equidistant in the z-direction. This is not always the case for HCP metals or any multi-basis crystal, as visualized in Fig. 2. Generally, $N_{plane}^{bulk}$ includes all atoms located inside a region with height equal to the minimum non-zero normal component of a lattice vector. In this work, we show that if the GB structure search considers only

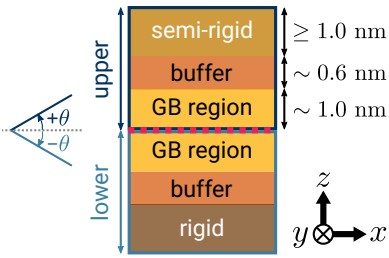

**Fig. 8 | Simulation cell setup for the Grand canonical Interface Predictor (GRIP).** The cell is oriented such that the $y$-axis is the tilt axis direction, the $x$-axis is the orthogonal in-plane direction, and the $z$-axis is the out-of-plane normal direction. Periodic boundary conditions are maintained in the grain boundary (GB) plane ($xy$-plane). GRIP begins by translating the upper crystal and removing atoms from the GB (magenta line). During dynamic sampling, atoms in the GB regions are free to move while atoms in the buffer and semi-rigid regions in the upper slab are constrained to move together, and atoms in the lower two regions are fixed. During relaxation, atoms in both buffer regions are free to move while the semi-rigid region is still constrained. For the $\gamma$-surface method, the upper slab is only allowed to translate as a whole before relaxation is applied.

those terminations of the surface by complete atomic layers, it is restricted to sampling states with $[n] = 0$ or $[n] = 0.5$ solely. Importantly, we demonstrate that this restriction misses lower-energy GB structures with intermediate values of $[n]$. A thorough search must consider all different atomic densities in the GB, as generalized by the framework presented here.

We implement this framework for handling the structural DOF and predicting GB phases in the open-source GRIP tool, written in Python with minimal dependencies (see Code availability). The algorithm rapidly samples the configurational space described by relative translations and different atomic densities and moves the system toward equilibrium. The relevant DOF—e.g., atomic density, reconstructions, temperature—are specified by the user in a single input file and the code exhaustively explores the GB phase space by sampling as many structures as possible in parallel. The energy calculations presented here use empirical IAPs to perform the dynamic sampling, but other techniques such as DFT can be used as well, as those calculations are decoupled from the structure optimization steps; however, the use of IAPs enables us to access low-angle GBs and larger reconstructions with thousands of atoms, as demonstrated here in simulations up to $3 \times 13$ reconstructions to validate the dislocation character. This methodology can thus take advantage of the increasing availability of computational resources and the advent of high-fidelity, machine-learned IAPs to enable quantum-accurate atomistic simulations of large systems with extended defects[55]. Advances in sampling and structure generation algorithms will further expand the diversity of results and the modular structure of the code enables different techniques to be easily plugged in. Of particular interest would be extensions to multicomponent systems, which could be handled using a Monte Carlo approach[23,24] for compositional DOF and would enable grand canonical sampling of GB structures in technologically relevant alloy chemistries.

## Methods
### GB structure search
We perform atomic-level optimization of GB structures using the open-source, Python-based GRand canonical Interface Predictor (GRIP) code (see Code availability), which rigorously explores structural DOF through dynamic sampling. Bicrystal slabs can be automatically generated using the Atomic Simulation Environment (ASE)[56] library for orientations where all orthogonal directions contain integer indices. Alternatively, they may be supplied as external files, which we create in this study for $\alpha$-Ti using a combination of ASE and

Pymatgen[57]. Figure 8 shows the orientation of the simulation cell, such that the $y$-axis is the tilt axis direction, the $x$-axis is the orthogonal in-plane direction, and the $z$-axis is the out-of-plane normal direction. We ensure periodicity in the GB plane ($xy$-plane) and an integer multiple of the interplanar spacing that totals at least 3.5 nm in the $z$-direction for each slab to minimize cell size effects. We note the $z$-direction is aperiodic and there is only one GB in the middle of the simulation cell. While any size cell can be used in principle, for computational tractability in this high-throughput study, we choose to simulate only [0001], [1$\bar{1}$00], and [1$\bar{2}$10] STGBs where all Miller–Bravais indices for the plane and the in-plane $x$-direction are less than or equal to 15, resulting in 16, 40, and 94 STGBs for each of the tilt axes, respectively (150 total).

For an individual GB, each iteration of the algorithm has three stages. During the first stage, the initial configuration is created by uniformly sampling a specific set of GB DOF. Specifically, the algorithm randomly samples an $m \times n$ replication of the unit GB cell (here, up to $3 \times 3$), randomly translates the upper slab in the $xy$-plane, and removes a randomly chosen (from the user-specified interval) fraction of atoms from the GB. The simulation box size scales with the replications to maintain in-plane periodicity. To further increase the structural diversity of the initial GB configurations, we have implemented random swaps of atoms on crystal lattice sites and interstitial sites in the GB region. The algorithm identifies interstitial sites near the GB as the vertices of the Voronoi diagram of the GB region.

During the second stage, it performs dynamic sampling to optimize the GB structure consistent with the imposed DOF. In this study, we used standard finite-temperature MD simulations using the Large-scale Atomic/Molecular Massively Parallel Simulator (LAMMPS)[58] in the canonical ($NVT$) ensemble with a Langevin thermostat and a time step of 2 fs as our dynamic sampling technique. The temperature and duration of the MD are also randomly sampled based on the user-specified ranges and only the atoms in the GB region are allowed to move freely during the dynamic sampling phase. It is straightforward to substitute this MD optimization with other, more sophisticated sampling techniques implemented in LAMMPS or other codes. For this study, we choose a GB region of 1 nm thickness on each side, a temperature between 300 K and 1200 K, and a duration up to 0.6 ns. Finally, the temperature is quickly ramped down to 100 K for 2 ps. With some probability (here, 5%) the algorithm skips the dynamic sampling for one iteration and jumps to the third stage.

In the third stage, each GB structure following MD sampling is fully relaxed at 0 K using a conjugate gradient minimization scheme, where atoms in the GB and buffer regions can move freely while the semi-rigid region is constrained to move together. Here, we specify the buffer region to be 0.6 nm beyond each side of the GB region, and larger values lowered $E_{gb}$ by no more than 1%. The convergence criteria are $10^{-15}$ for relative energy ($dE/E$ in successive iterations) and $10^{-15}$ eV Å$^{-1}$ for forces, with a maximum of $10^5$ evaluations for each criterion. The algorithm repeats these stages on each processor independently until termination, saving each relaxed structure to disk and periodically deleting duplicates. Duplicates are defined as structures with the same value of $E_{gb}$ and $[n]$ to three decimal places, and the algorithm will keep the structure with a smaller reconstruction and relative translations.

For each relaxed structure, the GB energy, $E_{gb}$, is computed according to:

$$E_{gb} = \frac{E_{total}^{gb} - N_{total}^{gb} E_{coh}^{bulk}}{A_{plane}^{gb}} \quad (1)$$

where $E_{total}^{gb}$ and $N_{total}^{gb}$ are the total energy and number, respectively, of atoms in the GB and buffer regions, $E_{coh}^{bulk}$ is the cohesive energy per atom in bulk $\alpha$-Ti, and $A_{plane}^{gb}$ is the area of the GB plane. We also track

the fraction of atoms in one plane or GB atomic density, $[n]$, according to:

$$[n] = \frac{N_{\text{total}} \bmod N_{\text{plane}}^{\text{bulk}}}{N_{\text{plane}}^{\text{bulk}}} \in [0,1) \qquad (2)$$

where $N_{\text{total}}$ is the total number of atoms in the simulation cell and $N_{\text{plane}}^{\text{bulk}}$ is the number of atoms in one plane of the bulk structure. Previous calculations of $N_{\text{plane}}^{\text{bulk}}$ simply counted the number of atoms at a single $z$ value in the bulk[4,11]; however, due to the 2-atom basis of the HCP crystal structure, atoms associated with one plane may be offset in the $z$-direction, as we show in Fig. 2. Therefore, we calculate $N_{\text{plane}}^{\text{bulk}}$ as the number of atoms within a region equal to the minimum non-zero normal component of a lattice vector; in HCP $\alpha$-Ti, this is equivalent to the interplanar spacing of the hexagonal lattice ($d_{hkl}$) given by[59]:

$$\frac{1}{d_{hkl}^2} = \frac{4}{3}\left(\frac{h^2 + hk + k^2}{a^2}\right) + \left(\frac{l}{c}\right)^2 \qquad (3)$$

where $h$, $k$, and $l$ are the Miller indices, and $a$ and $c$ are the HCP lattice constants. We note this extended definition of $N_{\text{plane}}^{\text{bulk}}$ reduces to taking a planar slice for unary, single-basis systems like elemental BCC and FCC metals, consistent with previous studies[4,11].

We compare the results of structure optimization using two different interatomic potentials, an embedded-atom method (EAM) potential for Ti–Al from Zope and Mishin[41] and a modified embedded-atom method (MEAM) potential for Ti from Hennig, et al.[38] For each STGB and potential, we also optimize the structure using the $\gamma$-surface method[19] for comparison, using a $2 \times 4$ replication of the same bicrystals and translating the top slab in increments of 0.025 nm in the $x$- and $y$-directions prior to a conjugate gradient energy minimization.

### High-temperature MD simulations
To study GB phase stability and transitions, we perform high-temperature MD simulations using methods adapted from previous work[4]. Briefly, we replicate the optimized GB structures in the $x$- and $y$-directions until the simulation cell is around 10 nm in the $x$-direction and 3 nm in the $y$-direction along the tilt axis. We freeze the bottom 1 nm layer of atoms and constrain the top 1 nm layer to be semi-rigid throughout the simulation (up to 20 ns). We use periodic boundary conditions (PBCs) in the $y$-direction and both PBCs and open surfaces with 1 nm of vacuum in the $x$-direction. We scan a range of temperatures between 600 K and 1200 K.

To induce a phase transition, we either insert additional Ti atoms at interstitial sites in the GB region or delete Ti atoms from a region near the top of the GB region. These MD simulations are performed in the canonical ($NVT$) ensemble between 600 K and 1200 K for up to 20 ns, using the MEAM potential and associated structures. For clarity of visualization, we relax all structures at 0 K using a conjugate gradient minimization scheme.

### DFT calculations
To validate select GB structures, we perform additional density functional theory (DFT) calculations using the Vienna Ab initio Simulation Package (VASP)[60–63] with projector augmented-wave potentials[64] and the generalized gradient approximation exchange correlation functional of Perdew, Burke, and Ernzerhof[65]. The semi-core $3p$ states are treated as valence states (`Ti_pv` potential). We use Monkhorst-Pack[66] **k**-point grids with a density of 5000 **k** points per reciprocal atom and apply Methfessel–Paxton smearing[67] with a width of 0.1 eV. The plane wave cutoff energy is 500 eV and the convergence criteria are set at $10^{-5}$ eV for energy and 0.02 eV Å$^{-1}$ for forces. We create the input structure by extracting a section near the GB region of the optimized structure from GRIP of approximately 4.5 nm in thickness (200–300 atoms) and adding 1 nm of vacuum on top. The axes are

rescaled to match equilibrium DFT values and atomic positions are fully relaxed while the cell shape and volume are fixed to maintain stresses in the GB plane. The energy of the GB is computed as the difference in total energy of a structure with the GB and a second bulk structure in the same orientation with the same number of atoms and vacuum but without a GB, divided by the planar area.

## Data availability
The data that support the findings of this study, including input and relaxed structures and the scripts used to generate them, are available in Zenodo under accession code https://doi.org/10.5281/zenodo.12590125[69]. The data used to generate the plots in this study are provided in the Source Data file. Other data are available from the corresponding authors upon request. Source data are provided with this paper.

## Code availability
The GRand canonical Interface Predictor (GRIP) tool that implements the GB structure optimization algorithm described here can be found at https://github.com/enze-chen/grip. The specific version used in this study is available in Zenodo under the same accession code[69].

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

## Acknowledgements
This work was performed under the auspices of the U.S. Department of Energy (DOE) by the Lawrence Livermore National Laboratory (LLNL) under Contract No. DE-AC52-07NA27344. Computing support for this work comes from the LLNL Institutional Computing Grand Challenge program and Bridges-2 at Pittsburgh Supercomputing Center through allocation DMR110087 from the Advanced Cyberinfrastructure Coordination Ecosystem: Services & Support (ACCESS) program. T.F. acknowledges support from the U.S. DOE, Office of Science under an Office of Fusion Energy Sciences Early Career Award. E.C. and M.A. acknowledge funding through the DOE, Office of Science, Office of Basic Energy Sciences, Materials Sciences and Engineering Division, under Contract No. DE-AC02-05-CH11231 within the Materials Project program (KC23MP). E.C. also acknowledges a fellowship through the National Science Foundation Graduate Research Fellowship Program under Grant No. DGE-2146752 and support from the Computational Chemistry & Materials Science Institute at LLNL. Part of this work was funded by the Laboratory Directed Research and Development program at LLNL under projects with a tracking code 22-ERD-002. Part of this work was also supported by the U.S. DOE, Office of Energy Efficiency and Renewable Energy, Hydrogen and Fuel Cell Technologies Office, through the Hydrogen Storage Materials Advanced Research Consortium (HyMARC). The authors thank Tomas Oppelstrup and Daryl Chrzan for helpful discussions. All figures are produced using matplotlib[68] and OVITO[40].

## Author contributions
E.C. and T.F. designed the study which was initially supervised by T.W.H. and B.C.W.; E.C. and T.F. designed the GRIP algorithm. E.C. performed the simulations and analyzed the results. E.C. and T.F. drafted the manuscript with feedback from M.A. All authors discussed the results and contributed to the writing of the manuscript.

## Competing interests
The authors declare no competing interests.
