## [Peer Review File · Nature Communications]

Grand canonically optimized grain boundary phases in hexagonal close-packed titaniumREVIEWER COMMENTS

Reviewer #1 (Remarks to the Author):

The manuscript reports the Grand canonically optimized grain boundary phases in hexagonal close-packed titanium. The manuscript is clearly written and contains some new results about GBs in HCP-structured materials with respect to the GB optimization. However, the following concerns must be addressed before this work is accepted for publication in NC.

(1) it seems this work is an example of GB optimization method application published by T. Frolov et al (Structural phase transformations in metallic grain boundaries. Nat. Commun. 4, 1-7 (2013)). In this reviewer's opinion, the authors must clearly address what's the difference in terms of methods and highlights between the present study and the methods reported in Nat. Commun. 4, 1-7 (2013).

(2) In Nat. Commun. 4, 1-7 (2013), the authors also reported another methods of driving GB phase transformation, i.e., by letting the defects gradually diffuse into GB from the side surface of GB model. So, does it still work for HCP GBs?

(3) It should be noted that this work does not mention also other methods at all when it comes to the GB optimization approach review, such as works due to MA Tschopp (RBT+atom removal), MJ Demkowicz (Vacancy loading) et al. In this reviewer's opinions, GCO only helps us to generate a larger unquilibrium GB structure space and such space may be larger than those created by those approaches by MA Tschopp, MJ Demkowicz et al. In essence, the ultimate goal of these methods are the same, keeping looking for the ground state of GBs.

(4) When the temperature is lifted and MD runs for some time, the system will be quenched to 0K, how does the temperature decrease rate affect the energy and structure of a GB state?

(5) P7, The $n = 0$ phase does not require insertion or removal of atoms and is metastable at 0K, how to judge $n=0$ is metastable?

(6) It is suggested that SUs should be marked in Fig.6

(7) P8: is it appropriate to use the term "the microscopic descriptor n " to say "properly captures all possible distinct GB configurations"? does the author mean that temperature is not important? If we say so, how does the temperature plays the role ?

(8) P 14, lines 240-241, a ref should be added.

(9) P14, lines 260-262, This statement requires caution and there must be some other MD simulations about the GB or interface. The dislocation network indeed changes upon the point defects introduction. It is a natural results of doing so.

Reviewer #2 (Remarks to the Author):

This is a solid work which provides a general tool (GRIP) for atomistic simulations of grain boundaries in a broad range of materials. Such a versatile tool facilitates understanding of both GB structures and energetics and is of a great significance for the materials modeling community. The manuscript is very well written and I recommend for publication provided the authors address the following questions:

- In Fig. 1(b), the data is fitted with a Gaussian distribution. Neither the data nor the whole process of GB ground-state search warrant in my opinion a normal distribution. Could the authors provide arguments for this?

- The probability peak in Fig. 1(b) is located close to the phase transition temperature in Ti. Is it a coincidence? How does it look like in other metals which were studied, for example, in W which does not show any phase transition and has a much higher melting point? Would there be a general range of nominal temperatures (T/T_{melt}) where the most stable GB phases are obtained with highest and fastest success rate?

- I find Fig. S10 in the Supplementary Note 1 very important and revealing and I would consider including it in the main text.

Firstly, it shows large differences between the MEAM and EAM predictions. I understand that these potentials are meant here as model potentials, but it should be still communicated that the differences between different models may be much larger than differences between the various possible metastable GB phases. Of course, only accurate and reliable potentials can be used for comparison with experimental observations. For instance, the GB structure in Fig. 3(g) predicted by MEAM looks very suspicious to me and it would be desirable to validate it using DFT.

Secondly, the large differences between GRIP and the gamma-surface sampling (sharp peaks) in panels (b) and (c) are worrying. I understand the different definition of planar terminations, but in the end there is just one definition of GB energy. If the gamma-sampling was used on the "easy" termination, one should arrive at the low energies as GRIP. Finally, GRIP seems to find low-energy GB phases mainly for the [0001] tilt GBs in Ti. Is there any explanation for that? I suppose the likelihood of more metastable structures will be linked to interatomic bonding (less in simple close-packed metals, more in materials with pronounced directional bonds).

Reviewer #2 (Remarks on code availability):

The provided URL does not exist. The authors should make it available.

Reviewer #3 (Remarks to the Author):

In this paper, the authors build upon their prior work on grain boundary phase transitions in FCC and BCC metals (two papers also published in Nat. Comm.) by considering grain boundary optimization and phase transitions for symmetric tilt grain boundaries in HCP Titanium. Whereas the emphasis of prior work on grain boundary phase transitions has been on high angle grain boundaries, the authors report the observation of a phase transition in low angle grain boundaries which involves a change in dislocation character via point defect induced dislocation pairing and un-pairing transitions. This finding is significant for its potential importance to modeling and understanding radiation damage effects in metals and alloys. Finally, the authors provide their code for an optimization method that

will be valuable to the community.

The paper is well written and the methodology appears to be sound. Several comments are listed below.

1. The novelty of the observed phase transition in low angle boundaries feels overstated. On Line 211, the authors state that “While low-angle GB phase transformations due to solutes and temperature have been previously reported... the questions of transition states and the role of intrinsic point defects have not been investigated”. As one counterexample, Szlufarska et. al (DOI:10.1038/s41598-018-21433-7) observed interstitial and vacancy induced changes in the dislocation character of low angle twist GBs in SiC. Granted that the above example is for a nonmetal, the authors should still cite this work and double check the literature for other reports of radiation induced change in dislocation structure in low angle grain boundaries.

2. I cannot find methodology for certain initial stages of grain boundary creation such as creation of periodic systems from coincident site lattice geometry. The authors should describe how simulation box sizes for symmetric tilt GBs in HCP Ti were determined. In addition, please clarify the additional challenges that exist for HCP vs. FCC/BCC. The challenge of the multi-lattice was already mentioned, but there are also additional challenges arising from an irrational c/a ratio – this makes periodicity normal to the GB plane challenging to enforce. Can this optimization method be generalized to systems with periodic boundary conditions normal to the GB plane which might be more useful for certain types of simulations?

Minor comments:

1. Figure S10: I see that around 40 degrees in (a) there is a case where Wang 2012 (EAM) finds a deeper minimum than GRIP (EAM). Are there cases where GRIP misses structures compared to prior work with the same potential and, if so, what would be possible causes for the discrepancy?

2. Line 20-22, last sentence of first paragraph: awkward wording, “structure-property relationships... have a profound influence on an array of phenomena”. The subject and object usage is inappropriate.

Reviewer #4 (Remarks to the Author):

Note: All Reviewer comments are in **blue** and Author replies are in **black**.

Reviewer #1 (Remarks to the Author):

The manuscript reports the Grand canonically optimized grain boundary phases in hexagonal close-packed titanium. The manuscript is clearly written and contains some new results about GBs in HCP-structured materials with respect to the GB optimization. However, the following concerns must be addressed before this work is accepted for publication in NC.

(1) it seems this work is an example of GB optimization method application published by T. Frolov et al (Structural phase transformations in metallic grain boundaries.Nat.Commun.4,1-7(2013)). In this reviewer's opinion, the authors must clearly address what's the difference in terms of methods and highlights between the present study and the methods reported in Nat.Commun.4,1-7(2013).

Author reply: We appreciate the reviewer's request for clarification and this opportunity to highlight the distinctions and advances realized in the present work compared to the approach used in the 2013 paper. The current manuscript demonstrates the success of the open-source GGrand canonical Interface Predictor (GRIP) tool, which we developed from scratch and enables high-throughput and highly automated GB structure optimization across diverse crystal systems and chemistries. The tool automates the process of structure generation and dynamic sampling of the relevant degrees of freedom, which led to the discovery of the new phases presented in the manuscript.

In Frolov, et al. *Nat. Commun.* 4, 2013, use was made of regular MD simulations with open-surface boundary conditions to allow for equilibration of the structure and the excess number of atoms in the boundary. Use was also made of an unpublished and no-longer-available script developed by David Olmsted (a co-author of the 2013 manuscript). To reproduce the MD-simulated structures, the script required significant retooling and our experience was that it was difficult to apply toward performing robust structure prediction more generally. Regarding the MD approach used in the 2013 manuscript, the large system sizes and finite temperature simulations required by the approach are such that the methodology must be customized for each GB and may miss low-/ambient-temperature phases.

Ultimately, the lack of comprehensive sampling and the need for custom modifications to the computer scripts required in the previously published methodologies motivated the development of the new GRIP tool to automate the optimization based on a single input file. Using GRIP, we show that this enhanced methodology not only convincingly reproduces previous GB phases in metallic and covalently-bonded systems (Supplementary Fig. S1), but it can also predict new phases in HCP Ti (main text) and in the well-studied BCC W system (Supp. Fig. S2).

Changes to manuscript: We believe that the manuscript already highlights the distinctions in the present study, e.g., "Herein, we perform..." (pg. 3) and "As motivated in the Introduction..." (pg. 6).

To address the reviewer's comment, we have modified a sentence in the Results—GRIP section to say: "GRIP performs a more thorough sampling of structures and optimization parameters compared to regular MD simulations with open surfaces. The periodic boundary conditions allow for convenient calculation of GB energy and atomic density, thereby enabling robust, high-throughput optimization of large GB datasets."

Pages changed: 6

(2) In Nat. Commun. 4,1-7(2013), the authors also reported another methods of driving GB phase transformation, i.e., by letting the defects gradually diffuse into GB from the side surface of GB model. So, does it still work for HCP GBs?

Author reply: When GBs are properly annealed during a sufficiently long MD simulation such that the GB structure and the number of GB atoms are optimized, the open-surface diffusion approach should work for any material, including Ti. However, this approach only works at relatively high temperature (where both surface and GB diffusion are rapid) and as a result can potentially miss low-/ambient-temperature GB structures. It requires large simulation blocks and is not suitable for the automated, high-throughput optimization that was performed in this study, as we describe in our response to the previous comment above. We developed the GRIP tool to enable more robust and automated structure prediction of GB phases, as demonstrated in this work.

To address the reviewer's question, we have performed additional high-temperature (up to $T = 1200$ K), open-surface MD simulations, following the methodology already described in Methods—High-temperature MD, which was adapted from Frolov, et al. *Nat. Commun.* 4, 2013. When we simulate the metastable state in Fig. 6c, we find the ground-state GB phase to appear after quite some time (~ 70 ns), as seen below.

We emphasize that these simulations demonstrate *feasibility* of the open-surface method. However, the approach is not suitable for general GB structure prediction, as different GB phases require careful tuning with this approach, and adequate sampling to find the equilibrium structure takes longer for some boundaries than others, due to large nucleation barriers between metastable and stable states (see Winter, et al. *Phys. Rev. Lett.* 128, 2022).

Changes to manuscript: We have included the above image as Supplementary Figure S10. We have also added the following sentence to the Discussion: “Additionally, we use high-temperature MD simulations with open surfaces to demonstrate first-order structural transformation between the different GB phases (Supplementary Fig. S10).”

Pages changed: 16, 35

(3) It should be noted that this work does not mention also other methods at all when it comes to the GB optimization approach review, such as works due to MA Tschopp (RBT+atom removal), MJ Demkowicz (Vacancy loading) et al. In this reviewer's opinions, GCO only helps us to generate a larger unequilibrated GB structure space and such space may be larger than those created by those approaches by MA Tschopp, MJ Demkowicz et al. In essence, the ultimate goal of these methods are the same, keeping looking for the ground state of GBs.

Author reply: We were previously aware of these two relevant studies and include them in the revised manuscript. We also would like to clarify that the reason we chose to omit them in our original manuscript was due to our understanding that they can fail to achieve the goal of "looking for the ground state of GBs." In our opinion, they cannot be classified as robust GB structure prediction methods.

With respect to Tschopp and McDowell, *Philos. Mag.* 87, 2007, the rigid body translation (RBT) approach with atom deletion is too restrictive and will not find the ground-state split kite phase or the high-temperature filled kite phase reported in $\Sigma 5$ Cu GBs because those phases correspond to $[n]$ fractions of $7/15$ and $6/7$, respectively (see Frolov, et al. *Nat. Commun.* 4, 2013). For these symmetric GBs the approach by Tschopp and McDowell which deletes overlapping atoms would either not delete any atoms or delete a full plane of atoms, generating structures with $[n] = 0$ only.

On the other hand, the vacancy loading approach detailed by Yu and Demkowicz, *J. Mater. Sci.* 50, 2015 is capable of sampling all the relevant distinct atomic densities $[n]$; in fact, Fig. 2 in that paper points out how the Tschopp method is trapped in the $[n] = 0$ state. Notably, Yu and Demkowicz bias their sampling by only introducing vacancies at the locations of lowest vacancy formation energy. GBs loaded with defects in this way require significant structural relaxation and it is unclear if the non-equilibrium states can sufficiently rearrange on the very short MD time scale. While it has been shown that this approach, like RBT, can lead to a reduction in the GB energy, it has not been shown if this method can produce more complex GB phases like $[n] = 7/15$ and $[n] = 6/7$ detailed in other studies.

Finally, we want to point out that in the Introduction (L41–42), we already cite several methods that perform GB structure search, including ref. 23 that Tschopp is a coauthor on.

Changes to manuscript: In the Introduction, we have added citations to:

- M.A. Tschopp and D.L. McDowell. "Structures and energies of $\Sigma 3$ asymmetric tilt grain boundaries in copper and aluminium." *Philos. Mag.* 87, 2007.
- W.S. Yu and M.J. Demkowicz. "Non-coherent Cu grain boundaries driven by continuous vacancy loading." *J. Mater. Sci.* 50, 2015.

and listed the methodologies as alternative approaches for atomistic modeling of GBs.

Pages changed: 2

(4) When the temperature is lifted and MD runs for some time, the system will be quenched to 0K, how does the temperature decrease rate affect the energy and structure of a GB state?

Author reply: The quench rate has a large impact on the success rate of generating the low-energy state, which is well known from the simulated annealing method. Quenching a disordered, high-temperature structure very quickly will result in a high-energy “glassy” GB, while quenching very slowly will result in the lowest-energy state for a fixed number of atoms [*n*]; however, this is computationally inefficient and we do not pursue this approach.

Instead, if the equilibrium GB structure already forms during the finite temperature simulation stage, then we can use a rapid quench to obtain the same low-energy state, as the quench rate will have no effect. This is the strategy used in GRIP, where we focus on parallel dynamic sampling of simulation time and temperature, and some of these simulations yield the ordered equilibrium phase. This strategy is illustrated in Fig. 1, where we analyze the success rate of generating ground states as a function of these parameters. Finally, we add that if one desires, sampling the quench rate can also be included in GRIP in a straightforward way.

Changes to manuscript: In the Results–GRIP section, we have added a sentence: “While it is well known that the quench rate impacts the success rate of generating the low-energy state~\cite{vonalfthan_2006}, this dependence is largely obviated when the equilibrium GB structure forms during the finite temperature simulations, from which a rapid quench is expected to yield the low-energy state.”

Pages changed: 5

(5) P7, The $n = 0$ phase does not require insertion or removal of atoms and is metastable at 0K, how to judge $n=0$ is metastable?

Author reply: The two GB phases shown in Fig. 3b ($[n] = 0$) and 3c ($[n] = 0.75$) have energies 0.519 J/m^2 and 0.514 J/m^2 , respectively. The metastable phase has energy only 1% higher than the ground state and is mechanically stable at 0 K. To further confirm that these are distinct (meta)stable phases, we performed additional MD simulations at $T = 300 \text{ K}$, 600 K , and 1000 K for 20 ns with PBCs in the xy -plane and observed that both structures remained stable for the entire duration in all simulations. Shown below are representative results at $T = 600 \text{ K}$. We will mention these validation checks in the main text of the revised manuscript.

Also, as explained in the manuscript, at lower tilt angles ($\theta \leq 6.58^\circ$), the ground-state structures are composed of $\frac{1}{3}[11-20]$ dislocations. We expect this dislocation structure to be preserved by continuity at higher tilt angles, even if those phases become metastable, which is consistent with our results.

3b, $[n] = 0$, initial

3b, $[n] = 0$, final

3c, $[n] = 0.75$, initial

3c, $[n] = 0.75$, final

Changes to manuscript: We have added a sentence in this section to say: “We perform additional MD simulations at high temperature (up to $T = 1000 \text{ K}$) for 20 ns with periodic boundary conditions and observe that both structures remain stable for the entire duration.”

Pages changed: 9

(6) It is suggested that SUs should be marked in Fig.6

Author reply: Structural units (SUs) have been outlined in black in Fig. 6 as suggested.

Changes to manuscript: Fig. 6 has been modified and the caption updated to include SUs.

Pages changed: 12

(7) P8: is it appropriate to use the term “the microscopic descriptor n ” to say “properly captures all possible distinct GB configurations”? does the author mean that temperature is not important? If we say so, how does the temperature plays the role ?

Author reply: We agree that the phrasing may be improved to avoid misinterpretation. In the current manuscript, when the sentence is considered in full, it has a different meaning and says that sampling the entire range of $[n]$ allows us to access different GB structures, *including* those that *could* be generated using different surface terminations.

Our interpretation of the reviewer's comment is whether $[n]$ is the *only* descriptor required to capture *all possible* GB configurations in all GBs. We did not mean to suggest this, as it is not true. This is why GRIP also samples relative translations between the grains as well as simulation temperature, as these are additional important parameters that must be sampled to discover all possible GB phases. Phrased another way, sampling $[n]$ is necessary, but not sufficient to produce all GB phases.

Changes to manuscript: The sentence in question has been changed to: “The GRIP search for the $\{31-40\}[0001]$ GB shown in Figure 3d illustrates that sampling the microscopic descriptor $[n]$ allows GRIP to identify all relevant distinct GB configurations, even for orientations with two distinct surface terminations.”

Pages changed: 9

(8) P 14, lines 240-241, a ref should be added.

Author reply: We assume the reviewer is referring to the sentence "Effectively, this sequence of states and partial transformations illustrate the possibility of GB transformation-mediated creep." We have added two references for Coble creep and a nucleation rate-limited GB creep model recently reported by Dillon, et al. Acta Mater. 246, 2023.

Changes to manuscript: At the end of the sentence in question, we have added two citations to:

- R.L. Coble. J. Appl. Phys. 24, 1963.
- S.J. Dillon, E. Lang, S.C. Finkeldei, J. Ouyang, K. Hattar. Acta Mater. 246, 2023.

Pages changed: 14

(9) P14, lines 260-262, This statement requires caution and there must be some other MD simulations about the GB or interface. The dislocation network indeed changes upon the point defects introduction. It is a natural results of doing so.

Author reply: Following the suggestion of the reviewer, we have removed the sentence, "To the best of our knowledge, such transition states facilitating the change in the GB dislocation network topology in a pure metal by point defect absorption have not been previously reported."

Changes to manuscript: We have removed the sentence in question.

Pages changed: 15

We thank the reviewer for their detailed comments.

Reviewer #2 (Remarks to the Author):

This is a solid work which provides a general tool (GRIP) for atomistic simulations of grain boundaries in a broad range of materials. Such a versatile tool facilitates understanding of both GB structures and energetics and is of a great significance for the materials modeling community. The manuscript is very well written and I recommend for publication provided the authors address the following questions:

- In Fig. 1(b), the data is fitted with a Gaussian distribution. Neither the data nor the whole process of GB ground-state search warrant in my opinion a normal distribution. Could the authors provide arguments for this?

Author reply: We agree with the reviewer completely and have removed the Gaussian curve, which purely served as a guide to the eye.

Changes to manuscript: The Gaussian curve in Fig. 1b has been removed and the caption updated accordingly.

Pages changed: 5

- The probability peak in Fig. 1(b) is located close to the phase transition temperature in Ti. Is it a coincidence? How does it look like in other metals which were studied, for example, in W which does not show any phase transition and has a much higher melting point? Would there be a general range of nominal temperatures (T/T_{melt}) where the most stable GB phases are obtained with highest and fastest success rate?

Author reply: We appreciate this nuanced question, as it is fundamental to the success of the dynamic sampling strategy. Based on our experience with different boundaries and different materials, we believe that GB diffusivity is one of the key factors; and the proximity to the phase transition temperature in titanium is a coincidence. Atoms and introduced vacancies have to diffuse to allow for GB structure rearrangement, which would typically occur on the timescale of the MD simulations at around 0.5–0.6 of T_m . For the same material, low-angle boundaries would require higher temperatures than high-angle boundaries. In addition, a transformation of one GB structure to another requires nucleation and growth of a GB disconnection (i.e., GB phase junction) loop. This is a thermally activated process that depends on the boundary type and can have a large barrier, which we recently quantified for tungsten (Winter, et al. *Phys. Rev. Lett.* 128, 2022). At some point, increasing temperature becomes counterproductive because high temperature also promotes high-energy GB states. There also could be low-temperature GB states that are relevant to ambient conditions, that could simply be not stable at 0.6 T_m . For these reasons, we believe that a broad range of sample temperatures is required, and we do not think we can limit ourselves to any specific, potentially optimal temperature.

Changes to manuscript: No changes to the manuscript were made because we believe this is already explained in the Results—GRIP section (pg. 5–6).

- I find Fig. S10 in the Supplementary Note 1 very important and revealing and I would consider including it in the main text.

Author reply: We are grateful to the reviewer for carefully checking our supplemental figures and for this suggestion. We have carefully considered the comment and have decided to keep Fig. S10 in the SI, but we will modify the text to include many of the reviewer's points as they strengthen the study's conclusions. Our rationale is detailed in the following responses to the sub-parts of this comment.

Firstly, it shows large differences between the MEAM and EAM predictions. I understand that these potentials are meant here as model potentials, but it should be still communicated that the differences between different models may be much larger than differences between the various possible metastable GB phases. Of course, only accurate and reliable potentials can be used for comparison with experimental observations. For instance, the GB structure in Fig. 3(g) predicted by MEAM looks very suspicious to me and it would be desirable to validate it using DFT.

Author reply: We felt that exploring different interatomic potentials (IAPs) of Ti, namely MEAM and EAM, would make the study more complete and we wanted to showcase the flexibility of GRIP in handling different potential parameterizations. The variability in predictions of different IAPs is somewhat known and has been investigated in previous studies; however, despite their limitations, well-tuned IAPs are known to predict experimentally observed GB structures. We focus on the MEAM potential in the main text because it is more likely to be accurate as it reproduces experimentally observed GB structures, as discussed in Ref. 30 (M.S. Hooshmand, et al. [arXiv:2103.06194](https://arxiv.org/abs/2103.06194), 2021). To the reviewer's acute observation, we include an additional comment about the energy differences between the potentials with respect to the energy differences between GB phases, with reference to a recent study by Mahmood, et al. *Scripta Mater.* 242, 2024.

We would also like to address the reviewer's concern regarding the structure shown in Fig. 3g. As can be seen from the energy plot, this is a very high-energy structure, with E_{gb} almost twice as large as that of the ground state. We certainly do not expect to be able to observe this structure at finite temperatures. It is included in this figure and the discussion to illustrate the limitations of the approaches that do not sample the correct number of atoms at the boundary. We expect that DFT calculations would also predict a very high energy for this structure.

Changes to manuscript: In the Results section where MEAM and EAM are both discussed, we have added the sentence: "We caution that a reliable interatomic potential (IAP) is crucial for accurate structure prediction; moreover, energy differences between different IAPs may be much larger than the energy differences between distinct GB phases~\cite{mahmood_2024}."

Pages changed: 10–11

Secondly, the large differences between GRIP and the gamma-surface sampling (sharp peaks) in panels (b) and (c) are worrying. I understand the different definition of planar terminations, but in the end there is just one definition of GB energy. If the gamma-sampling was used on the "easy" termination, one should arrive at the low energies as GRIP.

Author reply: Indeed, we obtained the sharp peaks by enforcing perfect terminations for illustration and intentionally didn't allow sampling of the second termination (which is the "easy termination" for some orientations). We wanted to underscore the importance of this sampling, but this wasn't clearly articulated and was overly strict. To eliminate confusion in the revision, we have updated our γ -surface methodology to sample different terminations and it now matches the previously reported results. We apologize for this oversight. Nevertheless, Fig. S10 still highlights how GRIP samples both terminations automatically by exploring the relevant $[h]$ without explicitly considering different surface terminations, as it is a lower bound for the γ -surface results.

Changes to manuscript: Fig. S10 has been changed to the following image and the associated text has been updated in Supplementary Note 1.

Pages changed: 36

Finally, GRIP seems to find low-energy GB phases mainly for the [0001] tilt GBs in Ti. Is there any explanation for that? I suppose the likelihood of more metastable structures will be linked to interatomic bonding (less in simple close-packed metals, more in materials with pronounced directional bonds).

Author reply: First, we note that for all three tilt axes, we find the lowest-energy structure at $[n] = 0.5$ for certain GBs (Fig. 4 and Fig. 5). The reviewer is correct that for axes other than [0001], the same ground states can also be found by sampling different surface terminations. In this sense, GRIP seems to be particularly relevant to [0001] tilt boundaries for this particular system. We also agree with the reviewer that the explanation is likely to be related to local bonding, but we are not sure that more directional bonding is the only ingredient that will lead to multiple metastable structures, since even for simple central-force potentials it can happen that many different bulk polymorphs are close in energy, and in such cases a diversity of metastable GB structures may also occur. The high-throughput exploration of different materials and boundary types enabled by GRIP would allow us to investigate this question in the future (since we don't know a priori) and explain how bicystallography affects the phase behavior of GBs. The prospect of exploiting GRIP with machine-learned potentials for systems with complex bonding is particularly intriguing in this regard.

Changes to manuscript: In Supplementary Note 1, we have modified the text (also in light of the previous point) and added the following sentence: "As it appears that GB phases are more prevalent for certain tilt axes than others, the high-throughput exploration of different materials and boundary types enabled by GRIP would allow us to investigate how bicystallography affects the phase behavior of GBs."

Pages changed: 37

Reviewer #2 (Remarks on code availability):

The provided URL does not exist. The authors should make it available.

Author reply: We are wholly in support of open science and reproducibility, and as stated in the Code Availability section, the GRIP code will become publicly available on GitHub upon publication. Meanwhile, we have already shared a ZIP file of the GRIP code with the editorial office.

Changes to manuscript: No changes were made to the existing language in the Code Availability section.

We thank the reviewer for their detailed comments.

Reviewer #3 (Remarks to the Author):

In this paper, the authors build upon their prior work on grain boundary phase transitions in FCC and BCC metals (two papers also published in Nat. Comm.) by considering grain boundary optimization and phase transitions for symmetric tilt grain boundaries in HCP Titanium. Whereas the emphasis of prior work on grain boundary phase transitions has been on high angle grain boundaries, the authors report the observation of a phase transition in low angle grain boundaries which involves a change in dislocation character via point defect induced dislocation pairing and un-pairing transitions. This finding is significant for its potential importance to modeling and understanding radiation damage effects in metals and alloys. Finally, the authors provide their code for an optimization method that will be valuable to the community.

The paper is well written and the methodology appears to be sound. Several comments are listed below.

1. The novelty of the observed phase transition in low angle boundaries feels overstated. On Line 211, the authors state that “While low-angle GB phase transformations due to solutes and temperature have been previously reported... the questions of transition states and the role of intrinsic point defects have not been investigated”. As one counterexample, Szlufarska et. al (DOI:10.1038/s41598-018-21433-7) observed interstitial and vacancy induced changes in the dislocation character of low angle twist GBs in SiC. Granted that the above example is for a nonmetal, the authors should still cite this work and double check the literature for other reports of radiation induced change in dislocation structure in low angle grain boundaries.

Author reply: We thank the reviewer for pointing us to the work by Jiang and Szlufarska, *Sci. Rep.* 8, 2018. We include a citation and amend the statement regarding a lack of investigation into point defect interactions in the revised manuscript. In the present study, one key difference is that we precisely characterize the metastable phase using the GB atomic density $[n]$ and show how the transformed region is the optimized, equilibrium structure for that particular value of $[n]$. While Jiang and Szlufarska show the change in dislocation character with point defect absorption, they do not, to the best of our understanding, show the connection to *distinct, equilibrium* GB phases.

At the reviewer’s suggestion, we have performed an extensive review of the literature and found only one other study by Martínez and Caro, *Phys. Rev. B* 86, 2012, that discussed a change in the dislocation network structure in a 2° $\{111\}$ twist boundary in FCC Cu upon vacancy accumulation. Like the study by Jiang and Szlufarska, they observe the transformation to begin at dislocation intersections that changes the stacking fault areas in the twist boundary. The distinction between these findings and those we are emphasizing in the present work relates to the observation of transformations between metastable GB phases. Here we demonstrate two phases composed of different dislocations and an equilibrium nucleus (Fig. 7c). The structures reported in the previous studies are not ordered and appear to be transient rather than related to stable and metastable phases and the nuclei involved in transitions between them.

Changes to manuscript: We have added citations in the section to:

- H. Jiang and I. Szlufarska. “Small-Angle Twist Grain Boundaries as Sinks for Point Defects.” *Sci. Rep.* 8, 2018.
- E. Martínez and A. Caro. “Atomistic modeling of long-term evolution of twist boundaries under vacancy supersaturation.” *Phys. Rev. B* 86, 2012.

and changed the sentence to say: “While low-angle GB phase transformations due to solutes and temperature have been previously reported by experimental observations and simulations in a few metals~\cite{refs}, a clear connection between intrinsic point defects and distinct, equilibrium, low-angle GB phases remains missing.” Corresponding changes have been made to the Discussion section, paragraph 2, to include these studies.

Pages changed: 12, 15

2. I cannot find methodology for certain initial stages of grain boundary creation such as creation of periodic systems from coincident site lattice geometry. The authors should describe how simulation box sizes for symmetric tilt GBs in HCP Ti were determined. In addition, please clarify the additional challenges that exist for HCP vs. FCC/BCC. The challenge of the multi-lattice was already mentioned, but there are also additional challenges arising from an irrational c/a ratio – this makes periodicity normal to the GB plane challenging to enforce. Can this optimization method be generalized to systems with periodic boundary conditions normal to the GB plane which might be more useful for certain types of simulations?

Author reply: We thank the reviewer for raising these important points regarding HCP geometry. When designing the GRIP optimization algorithm, we intentionally tried to make it flexible with respect to the user inputs. For the present study, the top and bottom grains were generated using an external script that used the Pymatgen and Atomic Simulation Environment (ASE) libraries to enforce the proper HCP orientations while keeping the smallest possible in-plane periodicity along the x - and y -axes. We had planned to provide this script with the released data (specified in Data Availability). Alternatively, GRIP itself can generate the top and bottom grains using only the ASE library, where the current implementation is restricted to grains where the mutually orthogonal axes can be fully specified using integer indices, i.e., rational ratios, due to constraints in the ASE code.

With respect to simulation box size, that (potentially) changes with each iteration of the algorithm. Regardless of how the input grains are generated, they are joined together to form the “unit” GB structure, or 1×1 in-plane reconstruction. The algorithm then randomly samples different $m \times n$ reconstructions of the unit GB in each iteration to cover a wide range of densities (based on user specifications) and the simulation box scales commensurately. In principle, this can be as large as your compute resources will allow. In the z -direction, we find that grains with sufficient distance between the GB and box edges ($\sim 30 \text{ \AA}$) is enough for a proper optimization, though this may be changed in the input file as the situation requires.

Finally, with respect to periodicity, the GRIP code may be modified to support PBCs in the z -direction normal to the GB plane. Certainly, even if an irrational c/a ratio prevents perfect periodicity along the z -direction, one can still select a specific termination to cut the grain at before enforcing PBC, assuming the boundaries are sufficiently far apart. For the purposes of GB structure optimization, we find a single GB in the middle of the cell to work best. The complexity of the optimization procedure makes doing two at the same time infeasible and may not be consistent in capturing all degrees of freedom.

Changes to manuscript: We have added several sentences to the Methods and Data Availability sections to clarify these points.

Pages changed: 17–18, 21

Minor comments:

1. Figure S10: I see that around 40 degrees in (a) there is a case where Wang 2012 (EAM) finds a deeper minimum than GRIP (EAM). Are there cases where GRIP misses structures compared to prior work with the same potential and, if so, what would be possible causes for the discrepancy?

Author reply: We are grateful to the reviewer for carefully checking our supplemental figures and for this observation. We do not believe GRIP misses structures compared to prior work. Rather, the differences seem to be due to some inconsistencies that we can only speculate may stem from different implementations used in the work of Wang and Beyerlein, *Metall. Mater. Trans. A*, 43, 2012 (specifically Fig. 10 of their article). To elaborate further, when we compare our results for GB energies to those published in an earlier study from Bhatia and Solanki, *J. Appl. Phys.* 114, 2013 (green bars in the plot below), we find excellent agreement with our results (orange bars). By contrast, the GB energies reported by Wang and Beyerlein (purple bars) are slightly lower for $\{11\bar{2}4\}$ (38.5°) than those obtained here and by Bhatia and Solanki, while they are significantly higher for $\{11\bar{2}2\}$ and $\{11\bar{2}1\}$ GBs, at 58° and 73° , respectively. As the $\{11\bar{2}2\}$ and $\{11\bar{2}1\}$ twin boundaries are far more common, it is surprising to us that the GB energies reported by Wang and Beyerlein are so much higher, and we can only infer that the discrepancies could be related to a slightly different methodologies.

In light of all of this analysis, we believe our methodology is consistent and GRIP does not miss any structures compared to prior work with the γ -surface method. The structures sampled by the γ -surface method are always a subset of the structures sampled by GRIP. Unfortunately, the details of the GB structures from the previous studies were not provided and we are unable to investigate the origin of the discrepancies in reported GB energies further.

Changes to manuscript: In Supplementary Note 1, we have added the sentence: “We note in panel (b) that the results from Wang and Beyerlein~\cite{wang_2012} exhibit discrepancies in E_{gb} compared to our calculations for certain orientations (e.g., {11-24}, {11-22}, {11-21}), whereas the results reported by Bhatia and Solanki~\cite{bhatia_2013} show excellent agreement. At present, the origin of these discrepancies remains unclear.”

Pages changed: 36

2. Line 20-22, last sentence of first paragraph: awkward wording, “structure-property relationships... have a profound influence on an array of phenomena”. The subject and object usage is inappropriate.

Author reply: We agree with the reviewer and have changed the sentence to say, “*The GB phase transitions* are believed to have a profound influence on an array of phenomena, such as diffusion and GB migration in materials.”

Changes to manuscript: The sentence has been changed as described.

Pages changed: 2

We thank the reviewer for their detailed comments.

Reviewer #4 (Remarks to the Author):

Additional changes to manuscript

We wish to inform the editors and our reviewers that we have made the following additional changes. Notably, **none of the conclusions of the paper have changed**, even when taking these points into account, which we hope everyone will agree with.

Notationally, all instances of “ n ” have been changed to “[n]” with square brackets, to better align with literature convention in expressing GB excess quantities (J.W. Cahn, in *Interfacial Segregation*, 1979; T. Frolov and Y. Mishin, *Phys. Rev. B* 85, 2012). The definition of [n], given in Equation 2, has not changed.

We again thank all four reviewers for their helpful comments that have improved the quality of this work.

REVIEWERS' COMMENTS

Reviewer #1 (Remarks to the Author):

The authors have addressed all comments. I would like to recommend its publication to NC.

Reviewer #2 (Remarks to the Author):

The authors have answered all my comments. I recommend the paper for publication.

Reviewer #3 (Remarks to the Author):

The authors have sufficiently addressed all comments from this reviewer.

Reviewer #4 (Remarks to the Author):

The authors have satisfactorily addressed the initial comments and I recommend publication.